# The Role of the Plasminogen Activation System in Angioedema: Novel Insights on the Pathogenesis

**DOI:** 10.3390/jcm10030518

**Published:** 2021-02-01

**Authors:** Filomena Napolitano, Nunzia Montuori

**Affiliations:** Department of Translational Medical Sciences and Center for Basic and Clinical Immunology Research (CISI), University of Naples Federico II, 80135 Naples, Italy; filomena-napolitano88@hotmail.it

**Keywords:** angioedema, fibrinolysis, plasmin, uPAR, tPA

## Abstract

The main physiological functions of plasmin, the active form of its proenzyme plasminogen, are blood clot fibrinolysis and restoration of normal blood flow. The plasminogen activation (PA) system includes urokinase-type plasminogen activator (uPA), tissue-type PA (tPA), and two types of plasminogen activator inhibitors (PAI-1 and PAI-2). In addition to the regulation of fibrinolysis, the PA system plays an important role in other biological processes, which include degradation of extracellular matrix such as embryogenesis, cell migration, tissue remodeling, wound healing, angiogenesis, inflammation, and immune response. Recently, the link between PA system and angioedema has been a subject of scientific debate. Angioedema is defined as localized and self-limiting edema of subcutaneous and submucosal tissues, mediated by bradykinin and mast cell mediators. Different forms of angioedema are linked to uncontrolled activation of coagulation and fibrinolysis systems. Moreover, plasmin itself can induce a potentiation of bradykinin production with consequent swelling episodes. The number of studies investigating the PA system involvement in angioedema has grown in recent years, highlighting its relevance in etiopathogenesis. In this review, we present the components and diverse functions of the PA system in physiology and its importance in angioedema pathogenesis.

## 1. Introduction

Angioedema (AE) is usually defined as a recurrent, self-limiting edema localized and lasting for a few hours or until five days. Tissue swelling is a result of increased vascular permeability, caused by the release of vasoactive mediators, in the deeper layers of the dermal, subcutaneous, mucosal, or submucosal tissues [1,2]. The most common clinical manifestations include swelling of eyelids, lips, mouth, tongue, extremities, and genitalia; abdominal pain and life-threatening airway swelling can also occur [3].

AE may occur alone, as a manifestation of urticaria, or as a component of anaphylaxis [4]. AE attack can result from bradykinin overproduction and/or from inflammatory mediators released by mast cells’ degranulation, primarily histamine. *De novo* synthesized mast cell mediators such as leukotrienes, prostaglandins, cytokines. and chemokines are probably implicated, but further studies are needed [5].

The disorder may be hereditary or acquired. According to the European Academy of Allergy and Clinical Immunology (EAACI) [6], five different forms of hereditary angioedema (HAE) have been identified: HAE-1, due to C1 inhibitor (C1-INH) deficiency, characterized by low antigenic and functional C1-INH levels; HAE-2, due to C1-INH dysfunction, characterized by normal (or elevated) antigenic but low functional C1-INH levels [7,8]; HAE-FXII, characterized by mutation in the coagulation factor 12 (F12) gene [9]; HAE-ANGPT1 with mutation in the angiopoietin-1 gene [10]; HAE-PLG with mutation in the plasminogen gene [11]. Recently, Ariano A. et al. identified a new form of HAE associated with a myoferlin mutation [12].

The acquired forms of angioedema (AAE) are classified into four types: AAE due to C1-INH deficiency (AAE-C1-INH) on acquired basis; AAE bradykinin-mediated (angiotensin converting enzyme inhibitor-induced angioedema, ACEI-AE); acquired forms related to mast cell mediators (urticarial angioedema, anaphylactic angioedema); and idiopathic angioedema [6].

Since the pathophysiology of these AE forms is different, correct diagnosis and appropriate therapy are crucial.

In this review we aimed to summarize the recent advances in the AE pathogenesis, focusing on the role of the plasminogen activation (PA) system. We also discuss the potential of the PA system as a new molecular target for AE therapy development.

## 2. The PA System: Not Only a Fibrinolytic System

The plasminogen activation (PA) system is composed by a series of serine proteases, inhibitors, and several binding proteins, which together control the generation of the active serine protease plasmin [13].

For many years, the PA system was known as the system responsible for vascular fibrinolysis, but it is not exclusively involved in this context. In addition to fibrinolysis, the PA system has been suggested to play an important role, as a source of proteolytic activity, during several physiological and pathological processes such as tissue remodeling, cell migration, tumor invasion, angiogenesis, wound healing, embryogenesis, inflammation, and immune response [14,15,16].

To understand the involvement of the PA system in a wide variety of processes, principal features of plasminogen, its activators, inhibitors, cell surface receptors, and extracellular ligands are to be analyzed. Hence, the main components of the PA system and their functions are described below and represented in Figure 1.

The precursor of plasmin, plasminogen (92 kDa), is synthesized by the liver and released in extracellular fluids, including plasma [17]. Native Glu-plasminogen consists of a single-chain glycoprotein of 791 amino acids, which contains *N*-terminal peptide (NTP) characterized by Glu terminal amino acid, five kringle domains (K1–K5), and a *C*-terminal domain, containing the protease domain. In the presence of lower amounts of plasmin, NTP is cleaved and Lys-plasminogen (85 kDa) is formed. The cleavage at the Arg_561_-Val_562_ specific bond converts the Lys-plasminogen single-chain polypeptide into the two-chain plasmin, linked by two disulphide bonds between the heavy A chain (60 kDa) and the light B chain (25 kDa) [13,18,19].

Plasmin is a protease with a broad substrate repertoire. The active plasmin degrades fibrin into soluble peptides eliminating its excess from blood and tissues [20]. In addition to this canonical function, plasmin can also cleave basic structural components of the extracellular matrix (ECM), such as fibronectin, collagen type IV, laminin, and proteoglycans, and is able to activate matrix metalloprotease (MMP) zymogens, such as MMP3, MMP9, MMP12, and MMP13 [21,22]. Plasmin can also activate or release various growth factors, cytokines, and chemokines from ECM including transforming growth factor (TGF-β), vascular endothelial growth factor (VEGF), and fibroblasts growth factor (FGF) [23].

### 2.1. The Plasminogen Activators

The plasminogen activation is mediated by intrinsic activators, so called as they are intrinsic to the plasma, and extrinsic activators, which are extrinsic to the plasma (Figure 1) [24].

Intrinsic plasminogen activators are active coagulation factor XII (FXIIa), FXIa, and kallikrein, which represents the contact activation system, also known as plasma kallikrein-kinin system, involved in both the intrinsic coagulation cascades as well as in fibrinolysis [25,26,27,28,29]. The intrinsic activators play relevant roles in inflammatory responses. In fact, the tissue undergoing an inflammatory insult contains negatively charged surfaces that recruit and activate the components of the contact system [14].

Extrinsic activators are represented by two serine proteases, the tissue-type plasminogen activator (tPA) and the urokinase-type plasminogen activator (uPA). The two types of plasminogen activators are encoded by different genes. The gene that encodes for uPA is located in the terminal portion of chromosome 10, while that for the tPA is located in the pericentromeric portion of chromosome 8. They have different structural and functional traits: The tPA is a fibrin-dependent enzyme and is mainly involved in blood clot fibrinolysis [30], while uPA is a fibrin-independent enzyme and mediates pericellular proteolysis [31].

The tPA is synthesized and stored in endothelial cells as a single-chain zymogen. Endothelial cells release tPA into the blood upon stimulation by thrombin, histamine, bradykinin, or other molecules. Single-chain tPA is then converted into a two-chain form by cleavage at the Arg_275_-Ile_276_ specific bond. The tPA binds to fibrin with similar plasminogen high affinity through the finger-like domain and kringle 2 of heavy A chain. The light B chain contains the protease domain [32,33,34].

The uPA, or “urokinase”, is synthesized by leukocytes, macrophages, tumor cells, and fibroblasts and released as a single polypeptide chain glycosylated zymogen called pro-uPA (411 amino acids). The pro-uPA consists of three domains: at the *N*-terminus a growth factor domain (GFD) that shares homology with the epidermal growth factor (EGF), a kringle domain (KD), and a serine protease domain at the *C*-terminus. A linker region, located between the *N*-terminal and the *C*-terminal region, undergoes cleavage of the Lys_158_-IIe_159_ peptide bond to produce a double-chain form of uPA (54 kDa) linked via a disulfide bond. The cleavage of pro-uPA is mediated by different proteases that include plasmin itself, cathepsins, trypsin, kallikrein, and mast cell tryptase. The two-chain form of uPA can undergo a further proteolytic cleavage at the peptide bond between Lys_135_ and Lys_136_ to generate a catalytically active low-molecular weight form of uPA (31 kDa) containing the serine protease domain and an inactive amino-terminal fragment (ATF) that consists of the GFD and the KD [35]. The uPA (both ATF form and the two-chain form of uPA as well as pro-uPA) can interact with a high-affinity cell surface receptor (uPAR), amplifying its capacity to activate plasminogen into plasmin. Apart from fibrinolysis, the uPA/uPAR interaction can enhance a cascade of events triggering cell migration, invasion, angiogenesis, and inflammation (Figure 1) [36].

The uPAR (CD87) is a single polypeptide chain cysteine-rich glycoprotein composed of three homologous domains (DI, DII, and DIII) connected to the cell membrane by a glycosylphosphatidylinositol (GPI) anchor [36]. Regardless of whether uPAR concentrates uPA activity on the cell surface, it is an adhesion receptor, as it binds vitronectin (VN) via the somatomedin B domain, an abundant component of provisional extracellular matrix (ECM), thereby regulating intracellular signaling and epithelial mesenchymal transition (EMT) [37,38,39]. In addition, uPAR regulates cell adhesion, migration, and proliferation through interactions with integrins [40,41,42,43,44]. By interacting with *N*-formyl peptide receptors (FPRs), uPAR also mediates both uPA- and fMLF-dependent cell migration [45]. Interestingly, uPAR mediates the shift from tumor cell dormancy to proliferation through an extensive cross talk with epidermal growth factor receptor (EGFR) [46]. The linker region between DI and DII domains of uPAR is extremely sensitive to various proteases, including uPA itself. The proteolytic cleavage removes DI and generates a shorter uPAR form (DIIDIII-uPAR), unable to bind both uPA and VN and to associate to integrins, but when exposing the SRSRY sequence (corresponding to aa 88–92) at N-terminus still able to interact with FPRs [47]. Both full-length and cleaved uPAR can be released by the cell surface in soluble forms (suPAR), which are detectable in body fluids and considered as a marker of disease severity in cancer and other life-threatening disorders [48,49,50,51].

### 2.2. The Plasminogen Inhibitors

The activity of the PA system is regulated at different levels by members of serine protease inhibitor (serpin) superfamily. The plasminogen activation is modulated by plasminogen activator inhibitor type 1 and type 2 (PAI-1/SERPINE1 and PAI-2/SERPINB2, respectively) [52,53,54].

PAI-1 and PAI-2 are single-chain glycoproteins (approximatively 50 kDa) and are encoded by two distinct genes, located on chromosome 7 and chromosome 18, respectively. PAI-1 inhibits rapidly both uPA and tPA. PAI-2 also reacts with both uPA and tPA, but slowly as compared to PA1-1 [55]. Elevated levels of PAI-1 are associated with several conditions, including coronary disease, myocardial infarction, and diabetes [56,57]. Interestingly, elevated levels of PAI-1 promote tumor progression, whereas high levels of PAI-2 appear to decrease tumor growth and metastasis [58].

The plasmin activity is controlled by alpha-2-antiplasmin and alpha-2 macroglobulin. Alpha-2-antiplasmin is a single-chain glycoprotein with a molecular weight of 70 kD and acts as the primary inhibitor of plasmin in plasma. Alpha-2 macroglobulin is a relatively nonspecific inhibitor of fibrinolysis that inactivates not only plasmin, but also kallikrein, tPA, and uPA [59].

The zymogen plasminogen is converted to the active protease plasmin by intrinsic and extrinsic activators. Intrinsic activators are members of the intrinsic coagulation pathway consisting of FXII, FXI, and kallikrein. Extrinsic activators are the tissue-type plasminogen activator (tPA) and the urokinase-type plasminogen activator (uPA). The tPA is released by endothelial cells as pro-tPA, a single-chain zymogen, and is converted into its two-chain active form by proteolytic cleavage. The uPA is synthesized as a single-chain zymogen (pro-uPA). Single-chain uPA is activated when it is converted to two-chain by plasmin (or other proteases) or when it binds as a single-chain molecule to its cellular receptor (uPAR). The uPA/uPAR interaction amplifies its capacity to activate plasminogen and also enhances a cascade of events triggering cell migration, invasion, angiogenesis, and inflammation. Inhibition of the plasminogen system may occur at the level of the plasminogen activators by plasminogen activator inhibitors (PAI-1 and PAI-2) or at the level of plasmin by α2-antiplasmin. The active plasmin degrades fibrin into soluble peptides eliminating its excess from blood and tissues. Plasmin is also able to activate matrix metalloprotease (MMP) zymogens, which play a key role in the degradation of the extracellular matrix (ECM).

## 3. The PA System in Angioedema: A Key Interplay between Vascular Dysfunction, Edema Formation, and Inflammation

The pathogenesis of angioedema (AE) reflects an intricate relationship among the coagulation system, the contact system, and the fibrinolytic system. Uncontrolled activation of these pathways triggers endothelial cell activation, increased vascular permeability, edema, and inflammation. It is well established that the principal mediator of symptoms of AE attacks is the bradykinin peptide, but the cellular and molecular mechanisms that cause its overproduction are not completely clarified.

Bradykinin is a nonapeptide particularly important in blood pressure regulation and in inflammatory reactions [60]. In physiology, bradykinin is released from its precursor high-molecular-weight kininogen (HK) by kallikrein, which is activated by active coagulation factor XII (FXIIa). Kallikrein, in turn, can activate FXII to generate more FXIIa (positive feedback). The major inhibitor of kallikrein and FXIIa is C1 esterase inhibitor (C1-INH). All together these factors represent the contact system [61,62]. After its generation, bradykinin (and Lys-bradykinin generated by tissue kallikreins) interacts with two specific receptors, the kinin B1 and B2 receptors (kB1R and kB2R, respectively). In detail, kB1R is expressed by cells at sites of inflammation, while kB2R is constitutively present on the vascular endothelium [63]. Administration of specific inhibitors of kB2R constitutes a valid and successful therapeutic approach, suggesting that the interaction between bradykinin and kB2R is crucial in HAE pathogenesis [64].

The fibrinolytic system is associated at several levels with bradykinin-forming cascade, as shown in Figure 2. Firstly, kallikrein, FXIa, and FXIIa can convert plasminogen to plasmin [25,26,29]. Secondly, it was demonstrated that plasmin can cleave and activate FXII in vitro and that thrombolytic therapy consisting of plasminogen activation leads to FXII activation in vivo [65,66]. Thirdly, bradykinin is able to stimulate endothelial cells to release tPA that, in turn, promotes plasmin formation [67].

To complicate the picture of the intricate molecular network of the PA system is the multiple relationships with the complement system. The complement system, with host defense functions, includes three separate activation pathways, the classical, alternative, and lectin pathways. All three pathways proceed to the activation of C3, the principle opsonic protein of the complement cascade, and then continue to the generation of activated factors, such as C5a that have potent inflammatory activity [68]. Importantly, plasmin is able to activate the classic complement system or to act directly on C3 and C5, thus producing, respectively, C3a and C5a (Figure 2) [69,70]. C3a and C5a are member of anaphylatoxins’ family as induced mast cells’ degranulation upon interaction with specific receptors expressed on mast cells’ surface, which results in histamine release, vasodilatation, and increased vascular permeability [14]. Histamine is one of the disease mediators of allergic angioedema, the most common form of AE. Plasmin seems to play a major role in histamine release.

In addition to these considerations, different alterations of the PA system in patients with AE, detected by laboratory tests and genetic analysis, have been described in literature. These combined observations assign to the PA system an active role in the HAE pathophysiology and show that understanding how this pathway can be inhibited by drug treatments ideally could lead to more effective therapies.

### 3.1. Increased Fibrinolytic Activity Is Associated with the Pathogenesis of Hereditary Angioedema

The first data that correlate fibrinolytic system to AE pathophysiology date back to 1985, when Nilsson T. et al. showed a constantly fibrinolysis activation during acute attacks in patients with hereditary angioedema with C1-INH deficiency (C1-INH-HAE) [71].

Currently, the model of “plasminflammation” has been proposed as a key pathway driving HAE pathogenesis [63]. The term plasminflammation indicates the connection between plasminogen activation and unregulated production of the inflammatory mediator bradykinin. In order to find an effective therapeutic approach for AE, Maas C. has affirmed that it would be necessary to neutralize plasminflammation through selective inhibition of the molecular interactions between the PA system and contact factor [63].

Below, we report the known fibrinolytic system alterations occurring in some different forms of hereditary angioedema (HAE) and the central role that plasminogen activation cascade plays in each of them.

#### 3.1.1. Alterations of Fibrinolysis in Patients with Angioedema due to C1-INH Deficiency

The classic forms HAE-1 and HAE-2 occur as a consequence of mutations in the C1-INH gene, SERPING1. HAE-1 is caused by mutations leading to a quantitative defect of C1-INH, whereas HAE-2 consists of qualitative defects of C1-INH function [72]. C1 inhibitor plays important roles in the regulation of vascular permeability through inhibition of FXIIa and kallikrein, involved in bradykinin production [73]. Bradykinin is accepted to be the principle mediator of swelling attacks in HAE, but the specific events that culminate in its increased release are not completely clarified.

Several recent evidences have highlighted the central role of the PA system in the unregulated production of the inflammatory mediator bradykinin in HAE-1 and HAE-2 forms.

Firstly, in a study of C1-INH-HAE (HAE-1), several plasma parameters of fibrinolytic activity were modified during acute attacks and could be considered valid biomarkers of disease. In fact, laboratory measurements in patients with HAE-1 during acute attacks have shown an increase in plasmin/alpha-2 antiplasmin complex (PAP complex) and plasmin D-dimer (product deriving from the breakdown of the fibrin mesh by plasmin), both markers of fibrinolytic activation [74]. In another independent study, PAI-1 plasma levels decreased significantly during acute attacks, suggesting that the activity of plasminogen is not controlled and its upregulation can be connected with swelling episodes [75].

In a transcriptomic study of HAE-1, Castellano G. et al. identified genes specifically modulated during acute attacks. In particular, they showed that PLAUR, the gene encoding uPAR, was significantly upregulated in circulating blood cells from patients during acute attack. Moreover, when uPAR expressed on T cells was neutralized, bradykinin production was reduced in a dose-dependent manner [76].

Although these experimental and clinical observations demonstrate that uPAR is associated with the pathogenesis of HAE attacks, the molecular and cellular mechanisms still remain to be elucidated.

The molecular mechanism through which bradykinin generation occurs on endothelial cell surface has been clarified.

In particular, zinc ion-dependent binding of the components FXII and HK of the bradykinin cascade to the cell surface is mediated by globular C1q receptor (gC1qR) and bimolecular complexes of gC1qR/cytokeratin 1 and cytokeratin 1/uPAR. HK binds preferentially to gC1qR/cytokeratin 1, while FXII interacts mainly with uPAR in cytokeratin 1/uPAR complex [77,78,79]. When prekallikrein is bound to HK assembled to this multimolecular complex, prekallikrein is activated into kallikrein that digests HK to produce bradykinin [80,81]. Binding of HK to endothelial cells is realized via a site within D2 and D3 of uPAR [79], which is likely identical to the site binding of VN [37,82], and this cross talk is able to stimulate the secretion of cytokines and chemokines from monocytes [83]. The generated bradykinin, in turn, stimulates tPA release from endothelial cells, thus enhancing the plasminogen activation [67].

It is, therefore, plausible that the overexpressed uPAR in HAE-1/2 patients acts as an amplifier circuit for bradykinin production through two well-established mechanisms. First, membrane uPAR is implicated directly in bradykinin release through recruitment of the constituents of the bradykinin-forming cascade on endothelial cells surface. Second, overexpressed uPAR indirectly could be linked to bradykinin formation by inducing plasminogen activation into plasmin that leads to activation of HK from which bradykinin is enzymatically released. In fact, among the functions of plasmin is the ability to cleave and activate FXII, as kallikrein can [65].

A biotechnological approach performed by Marceu F. et al. has provided new evidences of PA system-mediated bradykinin production in HAE-1 and HAE-2 pathogenesis. They revealed that bradykinin is released faster and intensely during incubation of plasma samples from HAE patients with tPA. Besides, tPA first activates plasminogen and secondly triggers contact system via FXII, as demonstrated by the fact that a specific plasmin inhibitor, DX-1000, blocked tPA-induced bradykinin formation but had no effects against kallikrein system [84].

Important evidence highlighting the role of the PA system in HAE pathogenesis is the use of antifibrinolytic drugs for prophylaxis and management of acute attacks. Since C1-INH is known to inhibit fibrinolytic proteases plasmin and tPA, the therapeutic approach based on restoration of C1-INHlevels in the HAE-1 patients with administration of exogenous C1-INH is the main cause of the thromboembolic complications [85]. Moreover, a valid and effective therapeutic approach to reduce the severity and recurrence of HAE attacks consist of the antifibrinolytic agents such as tranexamic acid and epsilon-aminocaproic acid. The antifibrinolytic drugs exert their activity through binding to specific lysine binding sites on the kringle domains of plasminogen, thereby reducing the amount of plasmin generated and limiting the downstream effects of plasmin itself.

The therapeutic benefits arising from targeting the PA system support the active role of plasminogen activation in the pathogenesis of classical forms of HAE.

#### 3.1.2. Alterations of Fibrinolysis in Patients with Angioedema due to Mutation in F12 Gene

In contrast to classic forms HAE-1 and HAE-2, a new type of HAE with normal C1-INH levels connected with increased FXII activity in plasma was recently identified [9].

FXII, also named Hageman factor, is encoded by F12 gene as a single chain and subsequently processed to active two chains’ form (FXIIa). It is a multi-domain protein that consists of an N-terminal fibronectin type II domain followed by epidermal growth factor-like (EGF) domain, fibronectin type I, EGF2 domain, Kringle domain, and C-terminal serine protease domain with an additional proline-rich sequence unique to FXII [86]. FXII is considered not only the initiator of the contact activation system, but its physiological role is related to fibrinolysis. Starting from results of a study conducted by Braat E.A. et al., it has been suggested that FXII exerts fibrinolytic properties as a plasminogen activator in the presence of a potentiating surface [87].

Two identified missense variants (T309K, T309R) in F12 gene cause variation of protein sequence in the proline-rich domain that connects the surface-binding domain of FXII to its protease domain. At first, it has been proposed that the substitution with positively charged amino acids may enhance activation of FXII as a result of an increased capacity to bind to negatively charged surfaces. Further biochemical analyses have shown that F12 mutations are gain-of-function mutations and introduce new cleavage sites in FXII molecule.

Interestingly, the pathologic forms of FXII make it more susceptible to cleavage and activation by plasmin.

De Maat S. et al. proposed that plasmin is a natural activator of FXII, controls physiologic bradykinin production, and regulates vascular permeability. In HAE-FXII, plasmin drives production of the inflammatory peptide bradykinin by upregulation of mutated FXII highly sensitive to activation [88].

It is also of interest that the main abnormality in plasma samples of patients affected by HAE-FXII was a significant decrease of PAI-2 levels, whereas a statistically significant difference in PA1-1 levels was not seen [89]. Contrarily, low PAI-1 levels were previously found in patients with HAE-1 and HAE-2 [74,75].

These observations, taken together, allow us to speculate about several important things: (1) PAI-2 deficiency can lead to excessive plasmin activation; (2) the F12 mutation enhances plasmin-mediated activation of FXII; and (3) prekallikrein is converted by active FXII into kallikrein that cleaves HK to release bradykinin, thereby producing an auto-amplification loop that sustains the bradykinin generation.

#### 3.1.3. Alterations of Fibrinolysis in Patients with Angioedema due to Mutation in Plasminogen Gene

Recently, Bork K. et al. newly identified a mutation c.988A > G in the plasminogen gene in German patients’ families affected by HAE with normal C1-INHlevels [11]. The genetic abnormality in HAE-PLG patients consists of a missense mutation in exon 9 of the plasminogen gene that leads to an amino acid substitution of lysine by glutamic acid in position 330 (p.Lys330Glu) in the kringle 3 domain of the plasminogen protein.

Interestingly, the same mutation transmitted as an autosomal dominant trait has been found by various groups in other families from different countries [90,91,92].

At the present time, little data about the prevalence and penetrance of the plasminogen gene mutation are available, but its pathologic role is supported by two important evidences. First, the identified mutation strictly co-segregates with disease. Secondly, HAE-PLG patients respond well to treatment with antifibrinolytic agents such as tranexamic acid.

From a pathogenetic perspective, the engagement of mutant plasminogen is reported as a critical step in the disease mechanisms. Mutant plasminogen is more accessible to plasminogen activators (such as uPA and tPA), with subsequent excessive plasmin-mediated generation of vasoactive bradykinin peptide in affected tissues.

### 3.2. Increased Fibrinolytic Activity Is Associated with the Pathogenesis of Acquired Angioedema

Different evidences from biochemical studies and clinical observations have highlighted the role of fibrinolysis in acquired angioedema (AAE).

In 1994, Cugno M. et al. reported that the activation of fibrinolysis, analyzed by measuring the plasma levels of plasmin/alpha-2 antiplasmin complex (PAP), resulted in elevated and continuous AAE with C1-inhibitor deficiency (AAE-C1-INH-) patients. Moreover, the counterproof of the fact that a fibrinolytic system was involved during AAE-C1-INH- attacks was provided by the favorable effects of prophylactic treatment with tranexamic acid [93].

Another important aspect of the AAE pathogenesis that assigns a key role to the plasminogen activation is represented by pharmacological thrombolysis that can provoke as adverse reactions AE attack. These forms are generally treated as histamine-mediated forms of AAE [94]. The thrombolytic therapy consisting of plasminogen-activating agents is the main treatment of ischemic stroke, but can result in some complications. Maertins M. et al. illustrated a case report of a patient who developed AE after administration of tPA for ischemic stroke [95]. Hill M.D. et al. also found evidence of orolingual angioedema after tPA treatment for ischemic stroke in 5.1% of patients recruited in a prospective study [96]. From a molecular viewpoint, infused tPA converts plasminogen into plasmin, which results in the wanted fibrinolytic effects. The plasminogen activation, in addition to fibrinolysis, may activate the kinin systems, leading to bradykinin generation, as already described above. Importantly, plasmin could induce the production of anaphylatoxins C3a and C5a, which contribute to mast cell degranulation and histamine release [14].

The role of the complement system in the generation of immunologic and/or allergic disease is of particular interest. This new knowledge clarifies some fundamental molecular aspects of the link between the fibrinolytic system and the complement system in the AE pathogenesis. In fact, the cross talk between the fibrinolytic system and the complement system amplifies the inflammatory process with the release of mighty inflammation mediators. Furthermore, the axis consisting of plasmin/anaphylatoxins/histamine underlies vascular disorders and edema development in AE forms associated with allergic disease and urticaria.

In the following Table (Table 1) we have summarized the main correlations between the specific forms of AE and the dysfunctions of the PA system.

## 4. Conclusions

The plasminogen activation (PA) system is a well-characterized proteolytic system composed of the zymogen plasminogen, the active protease plasmin, the urokinase-type (uPA) and tissue-type (tPA) plasminogen activators, a plasma membrane-associated receptor (uPAR), and two inhibitors, plasminogen activator inhibitor 1 and 2 (PAI-1 and PAI-2).

The PA system is mainly known for its function as a key component of the fibrinolytic cascade and is crucial in the maintenance of balance between coagulation and fibrinolysis. In the last decades, the growing molecular knowledge of the PA system has amazed the scientific community for its involvement into the wide variety of physio-pathological processes. The pleotropic activity is due to the ability of plasmin to contact other molecular pathways and of components of the PA system to establish plasmin-independent interactions.

Angioedema (AE) is a clinical-pathological entity defined as edema in the deeper layers of the skin and mucosa, caused by extravasation of fluid from the vascular compartment to the extracellular space. Five types of hereditary angioedema (HAE) and four types of acquired angioedema (AAE) have been identified.

AE attacks are provoked by bradykinin and/or mast cell mediators, principally histamine. The molecular alterations are triggered by two events: dysregulation of the kallikrein pathway and/or inappropriate activation of the complement cascade. To orchestrate this complex framework seems to be the responsibility of the PA system, located at the interface between the kallikrein pathway and the complement system. Beyond the fibrinolytic activity, the PA system is involved in a positive feedback reaction from which pro-inflammatory protein fragment or peptides are released.

In this review, we correlated the multiple properties of the PA system to different inherited and acquired forms of AE, thus adding small pieces to the complicated puzzle of AE pathogenesis.

The first consideration that arises from this study is that the contact between the PA system and the kallikrein pathway generates an auto-amplification loop that provokes abnormal production of bioactive peptide bradykinin. In these complex molecular dynamics, plasmin is considered a natural trigger for proinflammatory bradykinin production, but at the same time bradykinin itself stimulates endothelial cells to release tPA and plasmin is formed by this mechanism.

Secondly, plasmin can modulate complement activation via different mechanisms, generating anaphylatoxins C3a and C5a. This mechanism drives mast cells’ degranulation, which results in histamine-mediated vasodilation and increase of vascular permeability.

The uPAR acts as an amplifier for bradykinin production by recruitment on the cell surface of HK and prekallikrein. Most probably, HK binding to uPAR could be inhibited by recently discovered small molecules targeting the binding to uPAR of VN, which shares the same binding site on the receptor [97].

In conclusion, the mechanisms underlying angioedema pathogenesis are very complicated due to the intricate cellular and molecular communication that occurs. The several scientific evidences described in this paper highlight the loop in which the PA system, the contact system that generates bradykinin, and the complement system are connected. These observations can open the way to the development and characterization of therapeutic agents that disconnect plasminogen activation from production of the disease mediators.

## Figures and Tables

**Figure 1 jcm-10-00518-f001:**
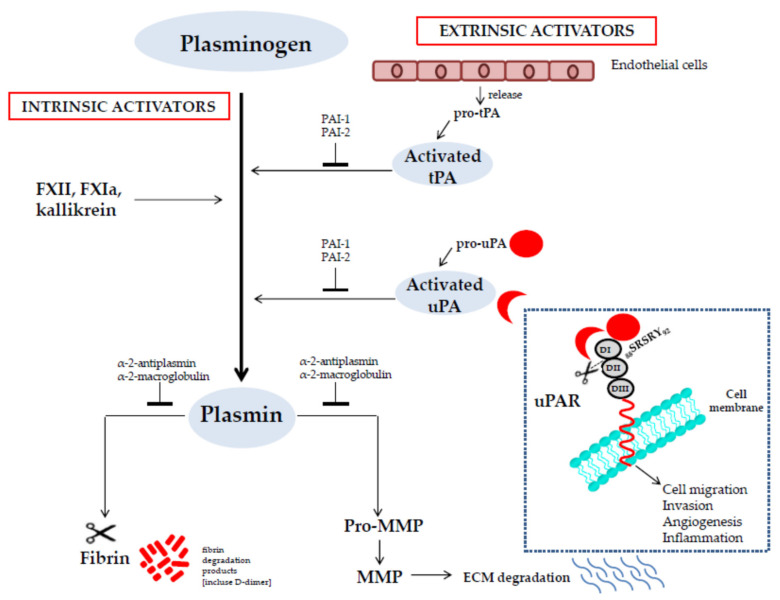
Schematic representation of the plasminogen activation system.

**Figure 2 jcm-10-00518-f002:**
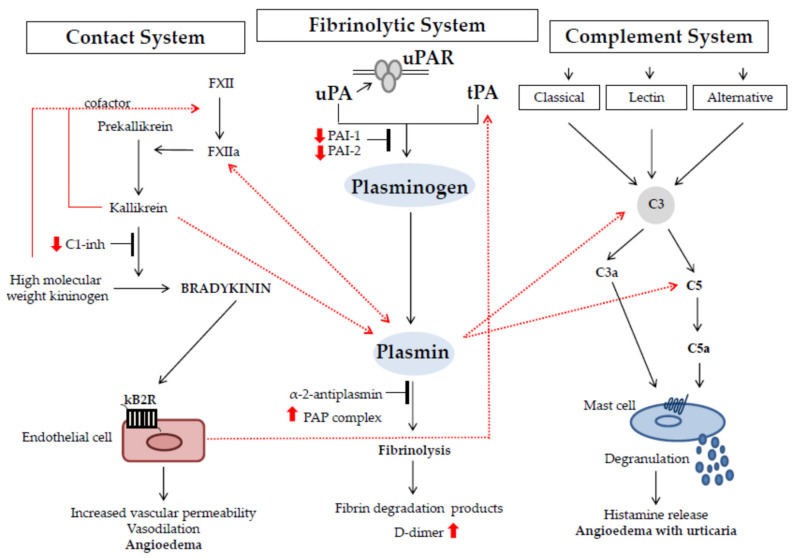
**Links between the fibrinolytic system, the contact system, and the coagulation system in AE pathogenesis.** The figure shows the complex multiple relationships that modulate the interplay of the molecules belonging to the fibrinolytic system. The red, dashed arrows indicate the contacts between the molecular pathways, while the red arrows indicate the increase or decrease of molecules in the plasma from patients affected by angioedema disease (AE). The interaction between the fibrinolytic system and the contact system leads to an auto-amplification loop that sustains the generation of bradykinin, the main mediator of AE attacks. Plasmin is also able to activate the classical pathway of complement and to cleave C3 and C5, generating C3a and C5a, respectively. The anaphylatoxins C3a and C5a induce mast cells’ degranulation and inflammatory mediators’ release. The most important preformed mediator is histamine, which mediates allergic forms of AE.

**Table 1 jcm-10-00518-t001:** Summary table of the fibrinolytic disorders in the different forms of angioedema.

Type of Angioedema Disease	Dysfunctions of the Plasminogen System
HAE-1 [C1 inh deficiency] and HAE-2 [C1-INH disfunction]	Increased PAP complex; Increased levels of D-dimer; Decreased PAI-1 level; PLAUR upregulation during acute attacks; tPA induced activation of plasminogen and kallikrein systems.
HAE-FXII [mutation in F12 gene]	Increased FXII sensitivity to activation by plasmin; Decreased PAI-2 level.
HAE-PLG [mutation in plasminogen gene]	Mutant plasminogen more accessible to uPA and tPA.
Acquired Angioedema (urtical Angioedema, anaphylactic Angioedema, idiopathic Angioedema)	Plasmin-induced C3a and C5a production, mast cells degranulation, and histamine release.

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
