# Peer review of "The Role of the Plasminogen Activation System in Angioedema: Novel Insights on the Pathogenesis"

_jcm, 2021, doi:10.3390/jcm10030518_

Round 1
Reviewer 1 Report
The contribution of the plasminogen activation system in the pathogenesis of angioedema has been recently recognised. Proteins involved in the system represent potential therapeutic targets. The authors reviewed sufficiently the evidence accumulated till now in regard with the function of this system and its its contribution to the pathogenesis of angioedema.
Reviewer 2 Report
In the following review, the authors thoroughly describe the pleiotropic function of the plasminogen activation system, with an emphasis on its role in the pathophysiology of angioedema. The manuscript is well written, and the various sections have a logical flow with extensive description of relevant recent discoveries. The information is also well summarized in the conclusion paragraph, which wraps up the manuscript nicely.
I only have a few minor comments:
- Is there a reason for using angioedema vs Angioedema (vs capital letter) differently in the text?
- Line 31. Angioedema does not necessarily last several days, and may have spontaneous resolution in only a few hours depending on the etiology
- Line 35. Replace “pain abdomen” with “abdominal pain”
- Line 143. The meaning of this sentence is not clear, particularly its ending (“and not only”…)
- Line 168. Replace “member” with “members”
- Lines 318-19. The meaning of this sentences is not clear.
- Line 345. Replace “trigger” with “triggers”
- Line 407. Replace “newly” with “new” or “newly identified a mutation…”
- Line 410. Replace “to lead to” with “that leads to”
- Line 458. Replace “histamine” with “histamine”
- Table 1. Replace “f12 gene” with “F12 gene”
Reviewer 3 Report
This is a very comprehensive and interesting review about the role of plasminogen activation system in angioedema, a pathway that appears to be fundamental to deepen the knowledge of this disease.
My major comments are about the intriduction, above all the clinical part.I will write my comments in order of lines.
In the abstract there is something missing. The authors write "angioedema is defined....mast cella mediators.". Then in the next period they talk about angioedema with C1 inhibitor deficiency, but they don't present this form, that is different from the forms mediated by mast cells. Probably a period about this part could clarify the topic for readers.
Introduction:
line 31: several is too generic in my opinion. better to write "until 5 days " or somthing like this.
Lines 31-32: is a result of increased permeability of what? increased vascular permeability?
Line 34: the zone around the eyes is called eyelid. Probably it is better swelling of eyelids, lips, etc...
Line 43: I think this paragraph needs a major revision. In the first part, the one about the hereditary forms, is based on the eaaci classification (bib. 6), in the second part they name the classification from HAWK (bib. 12). They are different. i think it is beter and less confoundign to use just one classificiation, or to clarify why the authors use both classfifications.
Line 43: in according to EAACI? or with EAACI? I think to is better.
Line 46: C1-INH is used for the forst time like acronyms, but there is not the clarification of the meaning.
Line 49: factor 12 gene is in italics
Line 51. Recently Ariano et al described a new form of hereditary angioedema linked to myoferlin mutation. It could be important to name it. Allergy, doi: 10.1111/all.14454
Line 52 and 53: this is a mistake. AAE doesn't refers to angioedema with acquired C1-INH eficinecy. This acronymus also refers to all the forms of recurrent angioedema with normal C1-INH, but not hereditary. In fact the authors name all the four forms in the next lines.
Figure 1: I think this figure is more easily readable if the legend contain the clarification of used acronyms.
Line 248: histamine is one of the disease mediator...it is not the only one.
Line 273-275: The authors say that the bib. 71 talks about Hereditary angioedema. Which time? It is important to clarify, because there are many types. It was for sure angioedema with C1-inh-deficiency, because in 1985 the other forms were unknown, but it is important to calrify for readesrs.
Line 283: alteration without '
Line 287,8: the authors use Cinh....sometimes they use C1-INH. it is important to use always the same acronym. The name of genes are in italics. These problems are present throught the paper. check it!
Line 299: congenital form is not so clear. better to write C1-INH-HAE
Line 396: PA1-1; correct with PAI-1
Line 399: to speculate about several.....
Line 400: Important things? a better word?
Line 401; a semicolon lacking between FXII and 3)
Tabel 1: C1 inh? better C1-INH; f12? in other part of the article the authors use F12. acquired angioedema?? which forms? this acronym include very different kinds of angioedema.
In the right part of the table increased levels of d-dimer is better in my opinion that increase of plasmin d-dimer.
Line 509-511: I think there is a large room for improvement in this conclusion
